# An algorithm for drug retrieval based on robot-grasping detection constraints and DDPG autonomous learning

**Xiaowen Zhang** **¹¤\*, Tiegang Lv²**

**1** Taiyuan Institute of Technology, Taiyuan, Shanxi, China, **2** Taiyuan Institute of Technology, Taiyuan, Shanxi, China

¤ No. 31 Xinlan Road, Jiancaoping District, Taiyuan City, Shanxi Province.
\* zhangxiaowen@tit.edu.cn

## Abstract

When the medicine-picking robot grasps drugs, its flexibility and accuracy in grasping detection mainly depend on the precision of visual guidance for the robot. The result of grasping detection directly determines whether the grasping task can be successfully completed. This study aims to enable a faster learning speed for the robot, reduce the search space for the grasping pose of the medicine-picking robot, and improve the grasping accuracy of the robot in unstructured environments. For this purpose, a self-learning DDPG grasping algorithm based on detection constraints is proposed and applied in automated pharmacy detection. The algorithm primarily consists of two steps. First, it extracts candidate grasping areas by analyzing the boundaries of the medicine. Second, with the aid of deep reinforcement learning, it inputs images with candidate grasping areas into an autonomous learning network, conducts adaptive noise exploration and perturbation in the search space, detects the optimal grasping point of the medicine from the image in real time, feeds it back to the medicine-fetching robot, adjusts the grasping pose through autonomous learning, and controls the robot to complete the training grasping. Experiments demonstrate that this method achieves a minimum of 15% improvement in grab detection accuracy compared with the four other grab detection methods. Within the confidence interval, it can achieve a grab success rate of 95%, which verifies the feasibility and effectiveness of this method.

## Introduction

Automated pharmacies not only innovate existing hospital pharmacy management but also accelerate work efficiency, reduce the workload of pharmacists, strictly supervise and control the circulation process of drugs, ensure medication safety, and conform to the characteristics of modern pharmacy management system safety, science, and efficiency [1]. The sorting and distribution of drugs can be completed when

**Data availability statement:** All relevant data are within the manuscript and its Supporting Information files.

**Funding:** The author(s) received no specific funding for this work.

**Competing interests:** The authors have declared that no competing interests exist.

dispensing robots are used to locate and grab drugs. The factors that affect the positioning, grasping, and detection accuracy of the drug retrieval robot include the end effector pose solved by the robot kinematics; the imaging model and calibration of various parameters of the vision system; and the target detection and grasping point pose. These factors directly affect the success rate of drug-grasping action. The most direct approach for ensuring accuracy in a medication-dispensing robot is to compensate for the time cost, which implies a competitive advantage over manual labor but is more difficult when ensuring a balance between accuracy and real-time performance. Furthermore, pharmacy automation requires robots to have autonomous learning capabilities for detecting the pose information of the target drug-grasping point during operation. This scheme can substantially improve the degree of pharmacy automation, such as drug grasping accuracy and real-time drug positioning.

When the drug retrieval robot performs grabbing tasks, the flexibility and accuracy of grasping detection depend mainly on the precision of the visual guidance provided by the robot. The results of grasping detection directly determine whether the grasping task can be successfully completed. Grasping detection collects environmental image information via visual sensors before processing is completed, and these tasks are pure machine vision problems. For the drug retrieval robot to complete grasping, the task must be analyzed and planned using relevant knowledge such as kinematics, control theory, and machine vision. However, when the dispensing robot operates according to the predetermined plan, the position of its end gripper always deviates, and errors ultimately accumulate. Therefore, the current position of the gripper must be judged in terms of its suitability for gripping. This judgment is subsequently used as a basis to guide the dispensing robot to adjust the gripping posture.

This study aims to enable a faster learning speed for robots, without relying on precise object and environmental information. The other aim is to quickly generate the grasping pose of the target drug based on the detection results of the drug detection algorithm, which satisfies the real-time requirements of the grasping operation. Thus, a grasping point detection algorithm based on DDPG autonomous learning is proposed and applied in automated pharmacies. This method, based on the results of object detection, generates grabbable areas and selects the location most likely to succeed in grabbing from these candidate areas. Then, it returns it to the medication retrieval robot. The process mainly consists of two steps. First, target objects are obtained from the target detection result image, and candidate grasping regions are extracted. These candidate regions are generated by analyzing the boundaries of the objects using edge detection operators. Then, these candidate regions are detected using deep reinforcement learning methods. By introducing adaptive noise perturbation into the exploration space, the optimal grasping box for the medicine is autonomously learned and detected from the image in real time. The graspable candidate regions are subsequently discriminated, which completes the mapping from the image to the grasping pose.

The rest of the paper is organized as follows. The second part organizes relevant literature, conducts theoretical comparisons with existing methods, and introduces traditional reinforcement learning methods. The third part proposes an improved

version of reinforcement learning based on exploration noise. The fourth part evaluates this method through extensive experiments and applies it to the grasping of a medicine-fetching robot. Finally, the fifth part summarizes the study.

## Related work

The grasping planning of robots involves finding a suitable pose from an infinite number of possibilities to achieve stable and effective grasping. Bohg [2] divided robot-grasping planning into two categories: analysis-based grasping planning and data-based grasping planning. Prior to the 21st century, research mainly used analysis-based grasping planning methods, which formulated grasping planning problems as optimization problems under multiple constraints [3,4]. Contact mathematical models for the grasping process are established, grasping constraints and objective functions are set, and stable grasping poses are obtained via optimization solutions. The analysis-based grasping planning method is computationally complex and assumes that the surface characteristics, weights and weight distributions, and geometric and physical models of the object are accurately known. Precise modeling of the grasped object is required during grasping. However, accurate models for all objects in complex real-world environments are impossible to establish, and real robots and sensor systems inevitably entail errors. These gaps limit the planning and application of analysis-based grasping in real-world environments. In addition, considering the complexity of the models of some robotic arms and the objects to be grasped, the computational complexity of solving the grasping pose is extremely high, and real-time grasping operation requirements are difficult to meet [5].

In recent years, with breakthroughs in deep learning in the field of image processing, researchers have applied deep learning to robot-grasping planning [6–8]. Deep networks can automatically learn high-quality grasping image features from a large amount of image data. Compared with the method of manually designing grasping features, the performance of newer models has been greatly improved, thereby advancing the grasping planning method for unknown objects. Deep learning-based grasping planning methods can predict feasible grasping poses in the images collected by sensors via neural networks and rank the candidate grasping poses by using evaluation indicators, after which the best grasping pose is determined [9]. For the grasping planning method of sampling candidate grasping poses, candidate grasping pose generation and optimal grasping pose selection are two key steps that directly affect the grasping efficiency and success rate. In 2015, Lenz [10] designed a two-step deep network cascade system that uses a relatively small deep learning framework to detect several high-quality grasping regions in RGB-D images. Once the detection results are output to a larger deep network, the optimal grasping pose can also be determined. In 2016, Johns [11] designed a convolutional neural network (CNN) architecture in which the input and output of the network are used as the evaluation scores of the depth image and all candidate grasping poses on the image, respectively. In 2018, Morrison [12] proposed a small CNN that does not rely on sliding windows or bounding boxes and instead directly generates grasping poses and grasping quality on depth image pixels. Compared with other networks, CNNs have fewer parameters of several orders of magnitude, which can speed up grasping pose detection. However, the greatest drawback of deep learning-based grasping planning methods is the difficulty in balancing detection accuracy and real-time performance.

The primary task of robot-grasping planning is to generate target-grasping candidate areas, which are generated on the basis of the external contour of the grasped drug. DRL networks are used for training to rapidly generate the optimal grasping area in these areas. In addition, DRL [13] refers to reinforcement learning combined with deep learning, which is an important branch of machine learning; it is a self-supervised learning method that has both the powerful perception ability of deep learning and the decision-making ability of reinforcement learning. Reinforcement learning [14] can describe and solve the problem of robot learning strategies, thereby maximizing returns or achieving specific goals during their interaction with the environment. Different reward functions are defined for different tasks, and the strategy network is optimized by continuously maximizing the reward value obtained at each step during the learning process.

The DDPG [15]is a commonly used DRL method for solving high-dimensional continuous state space tasks. The DDPG is based on the actor–critic learning framework and utilizes the deterministic policy gradient method, fully combining

the advantages of the Q-learning and policy gradient methods. DDPGs not only have unique policy characteristics of Q-learning that can fully utilize sampling data to reduce the number of robot trainings but also have the characteristics of the policy gradient method, which makes it applicable to high-dimensional continuous state space tasks. The algorithm framework is shown in Fig 1.

Reinforcement learning is an iterative process that solves two problems at each iteration. Given a policy evaluation function, the policy is updated on the basis of the value function. The control strategy μ is the grasping strategy of the robot, and the $Q$ network represents the action value function. In the reinforcement learning training phase, the control strategy obtains the current state $s_i$ and outputs the action instructions $a_i$ of the robot. After the robot executes the action, the state of the operated object changes. This changed state is denoted by $s_{t+1}$. The reward value $r_t$ is calculated on the basis of the reward function. The robot stores the abovementioned four attributes as a tuple $(s_t, a_t, s_{t+1}, r_t)$ in its experience pool. When the amount of data stored in the experience pool reaches a certain level, the $Q$ network is trained on uniformly sampled data from the pool. After training the $Q$ network, the deterministic policy gradient method updates the robot control strategy. The robot continuously repeats this process until the control strategy converges.

On the theoretical foundation of deep reinforcement learning, many researchers have begun to devote efforts to studying its application in robotic grasping detection. Zeng et al. [16] successfully completed the grasping and pushing actions of a robotic arm in an unstructured environment, in virtual settings and with real robots. In their study, they utilized observations from an RGB-D camera as input, integrated grasping and pushing strategies, and employed a Q-network to compute the maximum Q-values for these actions, which facilitated optimal action selection. Based on an improved DDPG algorithm. Dmitry Kalashnikov [17] from Google Brain adopted a scalable self-supervised learning method based on reinforcement learning, which is called QT-Opt. This method conducts grasping training on robots. During the training process, objects that are grasped receive a reward of 1, while objects that are not grasped receive a reward of 0. The collected data are used to train a deep reinforcement Q function, which evaluates the grasping position and orientation. Based on the evaluation values, the optimal grasping position and orientation are selected. After training 580,000 steps on robots, the success rate of grasping unknown objects reached 96%. To address the issue of the large number of training steps required by deep reinforcement learning algorithms, multiple robots are used to simultaneously train on grasping tasks to shorten training time.

**Fig 1. Block diagram of the DDPG algorithm.**

Breyer et al. [18] compared various robotic arm grasping methods based on reinforcement learning and proposed a deep reinforcement learning grasping method based on an action offset strategy. This method first uses an encoder–decoder network to reconstruct the depth image, which generates a depth prediction map containing embedded vectors to enhance the expressive power of features. Then, the embedded vectors are concatenated with the robotic arm state vectors, and a reinforcement learning policy network is used to predict the action offset. Finally, the robotic arm performs the corresponding grasping action based on the offset output by the network.

Zhu [19] proposed an experience pool sampling strategy based on reward value priority and a method based on adaptive target exploration domain. Experiments conducted in a simulated environment for robotic arm grasping training showed improvements in terms of time required for convergence, number of steps needed for convergence, and convergence stability.

Zhang [20] proposed a non-rational angle constraint strategy that dynamically evaluates the grasping effectiveness in different directions and imposes varying degrees of constraints on grasping actions in non-rational directions, which enhances the learning efficiency and grasping success rate of target grasping. By introducing the constraint factor in the non-rational angle constraint strategy, a grasping reward function with angle information is proposed to guide the grasping network to learn more accurate and detailed grasping strategies. To address the problem of sparse rewards during target grasping training, a grasping training method based on post-experience replay is adopted, which effectively improves the efficiency of target grasping training.

You Tianya [21] proposed a deep reinforcement learning algorithm based on an improved Actor–Critic framework, which utilizes pushing actions to change the state of the environment and provide operational space for grasping when necessary. In scenarios with dense target objects, the algorithm achieves varying degrees of improvement in grasping success rate, action efficiency, and task completion rate.

The grasping method based on deep reinforcement learning conducts grasping training through self-supervision, which continuously interacts with the environment to master stable grasping skills. This method does not require manually creating a large amount of labeled data to supervise model learning and exhibits good environmental adaptability. However, current grasping methods based on deep reinforcement learning often suffer from issues such as low model training efficiency and slow convergence speed. The literature [22] also points out that in engineering, the focus should be on providing higher autonomy for robotic systems. Therefore, more effective action strategies, learning methods, or reward mechanisms need to be introduced for faster grasping skills of the robotic arm.

## Proposed method

This section primarily introduces our proposed DDPG autonomous learning grasping detection method for medicine-fetching robots based on detection constraints. In this method, candidate grasping areas are generated using target detection results to enhance the accuracy of grasping poses. The boundaries of the generated targets are also calculated and analyzed. Under the constraints of detection results, an autonomous learning grasping strategy network is designed, which outputs a reference rectangular box for grasping targets. Through action function mapping, the grasping pose is obtained. An adaptive exploration noise strategy is designed for perturbation to achieve broader exploration and autonomous learning within the grasping space of the robot. The details are elaborated below.

### Candidate capture area generation

Given that this study primarily focuses on the grasping of boxed drugs, the chosen end-effector method is suction-based. The pose for grasping the drug is usually the center position of the medicine box. According to the YOLOv3 object detection algorithm output, the rectangular box of the drug encompasses the entire drug. However, some errors may exist between the output rectangular box and the actual boundary of the drug on the captured image. Candidate grasping areas are generated based on the boundary features of the drug to obtain an accurate grasping pose. An autonomous learning

network is used to detect the suggested areas, and the optimal position is finally selected and controlled for grasping. This process is represented mathematically as follows:

$$g = \max\left(N\left(r\right)\right) r \in R \qquad (1)$$

where $R$ represents the candidate grasping area generated after drug detection, $r$ is the possible grasping position within the candidate grasping area, $g$ is the optimal grasping pose, and $N$ denotes the mapping function corresponding to the autonomous learning network.

Based on the constraints of force closure and shape closure, a rectangular region is set up to more accurately describe the grasping pose of the end effector and the spatial position of the medicine. This area is represented by a 5-tuple vector, as shown in the Fig 2.

The captured area in the image is represented by a 5-tuple as $(x, y, w, h, \theta)$. Within the rectangular area, the long side of the red rectangular box indicates the width of the medicine box $w$, the long side of the blue rectangular box indicates the length of the medicine box $h$, $(x, y)$ corresponds to the coordinate value of the upper left corner of the positive direction rectangle of the medicine box, $\theta$ describes the direction of the rectangle, and the black box represents the ideal adsorption area of the end effector. From the results of drug detection, the $(x, y)$ coordinates corresponding to the points on the plane can be easily determined. By utilizing depth information to determine the z coordinate, the grasping area can be represented as $(x, y, z, w, h, \theta)$. This 6-tuple describes the candidate area for the medicine-picking robot to grasp the medicine.

The purpose of candidate grab regions is to determine the precise location of the medicine. It involves dividing the panoramic image into local regions, followed by further judgment to avoid redundant work. After the medicine detection is completed, the regions containing medicine are used as the search space for candidate regions. Generating candidate

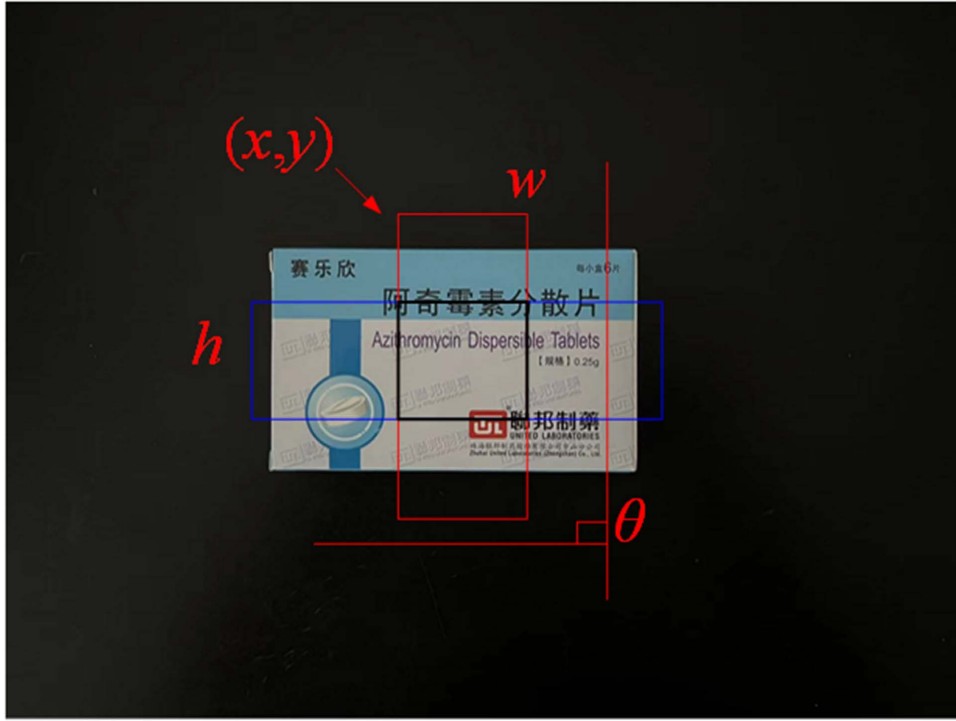

**Fig 2. Schematic diagram of the 5-tuple of the grab area.**

grab regions in the vicinity of the medicine detection rectangular box can significantly reduce computational workload and improve grab time. The optimal representation of the candidate region is when the rectangular box overlaps with the boundary box of the medicine, which can contain the complete medicine and eliminate the redundant background. This area can serve as the search space for the grasping pose of the medicine-picking robot. Based on prior experience, a set of potential grab regions is extracted at the boundaries of the drug to balance computational load and window traversal times.

First, the boundaries of the drugs in the image are extracted; then, candidate regions are generated along the boundary curves in a clockwise direction, as shown in Fig 3. Starting from the top left pixel on the curve, the coordinates of the top left corner of the candidate region box are determined at intervals of 10 pixels. The normal direction of the fitted line of the set of these 10 points is used as the angle between the candidate region and the horizontal direction. Combining relevant knowledge of the rotation matrix, the candidate region including the boundary can be obtained and the region size can be determined using Equation (2).

$$\left(\widehat{x}, \widehat{y}\right) = \begin{bmatrix} \cos\theta & \sin\theta \\ -\sin\theta & \cos\theta \end{bmatrix} (x, y)$$

(2)

## Network design of the self-learning DDPG capture strategy

Given that the geometric and physical information of the captured drugs is unknown, a specific strategy function or robot action library cannot be easily used to complete the robot-grasping task. The characteristics of the current drugs must be understood on the basis of sensor information to enable the drug retrieval robot to grasp drugs in unstructured environments. Autonomous learning actions for drug retrieval on the basis of these characteristics must also be planned accordingly. Drawing on the powerful image feature extraction ability of deep CNNs and combining it with reinforcement learning self-learning ability, a strategy network for drug retrieval robots to grasp drugs was designed in this research.

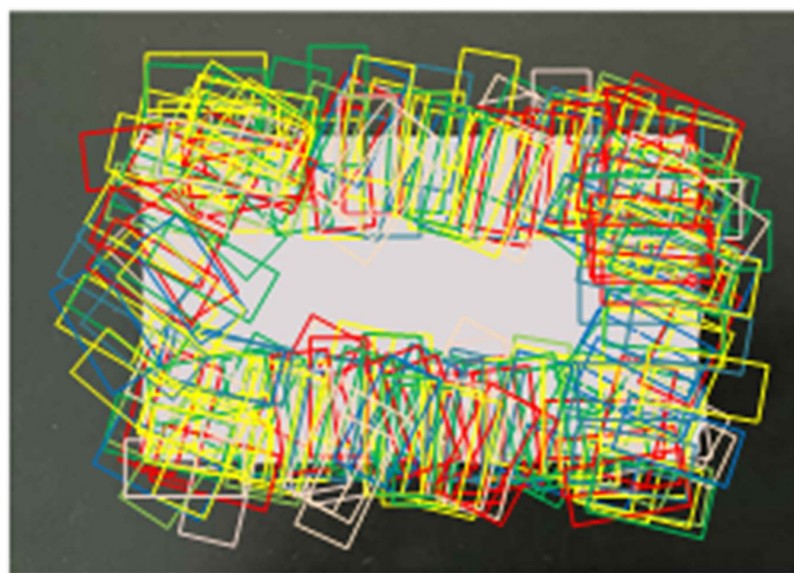

**Fig 3. Schematic diagram of candidate region generation.**

The input and output of the strategy network are initially clarified. The strategy network captured by the drug retrieval robot performs mapping from the image of the captured drug to the position and posture information captured by the robot. Therefore, the input of the strategy network can be regarded as the candidate region image containing the position, posture, and shape information of the drug after drug detection (i.e., the state of the Markov decision process). The pose of the robot during grasping is expressed as $g= (x,y,z,w,h,\theta)$. In terms of the selected adsorption type end effector in the experiment, $w$ and $h$ represent the width and length of the medicine box, respectively, and $\theta=90°=\pi/2$ is the angle in the $w$ and $h$ directions. The output is not the grasping pose of the robot but the pixel coordinates $(u, v)$ corresponding to the grasping position of the robot. Rather, the position of the grasping point is calculated via the depth value and three-dimensional coordinate conversion as follows:

$$
\begin{aligned}
z_{cam} &= \frac{d}{s} \\
x_{cam} &= \frac{(u-u_0)z}{f_x} \\
y_{cam} &= \frac{(v-v_0)z}{f_y}
\end{aligned}
$$

(3)

The abovementioned parameters are all camera-related parameters. After the position of the pixel point in the camera coordinate system is obtained, the position of the pixel point in the robot coordinate system is generated via a homogeneous transformation of the camera coordinates and the dispensing robot coordinate system.

$$
\begin{bmatrix} x_{robot} \\ y_{robot} \\ z_{robot} \\ 1 \end{bmatrix} = T_{robot}^{cam} \times \begin{bmatrix} x_{cam} \\ y_{cam} \\ z_{cam} \\ 1 \end{bmatrix}
$$

(4)

The position obtained in the robot coordinate system mentioned above is a theoretical value. Due to error factors such as lens distortion in the vision system, the calibration method from reference [23] was selected during camera calibration, which to some extent reduced the existence of errors.

Assume that the image size used is $256 \times 256$ and that $u$ and $v$ are within the interval [0,256]. During network construction, a simple mapping relationship $u' = u \times 128 + 128$ is used to map the value space of $u$ and $v$ to $[-1,1]$. $u'$ represents the actual pixel coordinate. The analysis proves that the output of the strategy network is the variable $(n_x, n_y, n_z, n_w, n_h, n_\theta)$ at $[-1,1]$.

According to the description of the capture actions for reference matrix boxes [24], each sampling point in the feature map is a captured reference rectangular box set at a certain scale and aspect ratio in the original image. The position of the grab rectangular box can be determined by the offset and the grab reference rectangular box. $(n_x, n_y, n_z, n_w, n_h, n_\theta)$ represents the geometric coordinate offset of the predicted grab rectangle obtained via regression from each rectangular box relative to the reference. Similarly, for each sampling point, corresponding to an $n$ reference grabbing rectangular box, the output predicted grabbing rectangular box's offset parameter dimension is $3n$. Therefore, the output layer is equivalent to two $1 \times 1$ convolution kernels, which convolve the transition layer in terms of effect. The self-learning DDPG grasping strategy network model used to achieve an end-to-end output is shown in Fig 4.

The size of the candidate capture area image is fixed at $256 \times 256$, which serves as the input for neural network training. The first few layers use a CNN for feature extraction, and the convolutional kernel size selected by the convolutional layer is 1*1. After a series of convolution operations, a feature map is obtained. The maximum pooling layer is used to reduce the image dimension after the main image features are extracted. The convolutional kernel selection of the maximum pooling layer is $2 \times 2$. The fully connected layer is subsequently used to obtain the intermediate transition layer. In the intermediate transition layer, in which the image feature and the corresponding variable outputs are considered, the parts for feature extraction and capture detection of the entire network, as well as the geometric parameters of the predicted

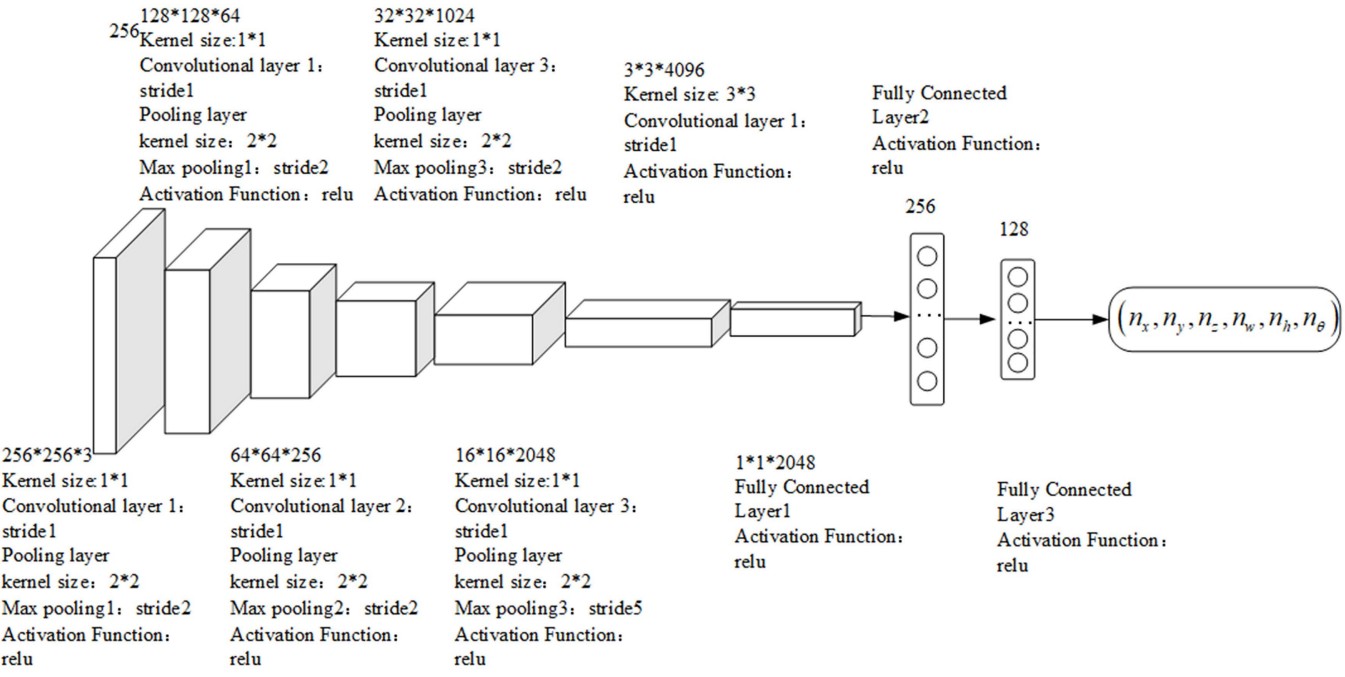

**Fig 4. Network structure diagram of the grasping strategy of the medicine-taking robot.**

capture rectangular box, are used to calculate the robot's actions. The activation functions of the convolutional layer and the first two hidden layers of the fully connected layer are given by ReLU functions. The $u$ function, with the final output layer taken as the *tanh* function, controls the output variables of the strategy $[-1,1]$ mentioned above.

### Mapping of the grasping action function of the medication retrieval robot

The neural network output of the drug retrieval robot's grasping strategy consists of six variables with a value range of $[-1,1]$. This strategy is represented by $(n_x, n_y, n_z, n_w, n_h, n_\theta)$, which corresponds to the position $(x, y)$, depth value $z$, width $w$, length $h$, and angle $\theta$ of the robot's end effector. Therefore, a mapping relationship between the pixel point area $(u_{max}, u_{min}, v_{max}, v_{min})$ of the target drug from $(n_x, n_y, n_z, n_w, n_h, n_\theta)$ to $(x, y, z, w, h, \theta)$ needs to be established, and the main task is to determine the relationship between $(n_x, n_y)$ and $(x, y)$. The $(n_x, n_y)$ output of the strategy network represents the relative position in the robot's workspace. If the robot's actions are limited to the candidate grasping area, then only $(n_x, n_y)$ needs to be converted to the position within the relative candidate area.

A coordinate system *ouv* is established with the center of the border of the candidate grasping area as the origin, and $(n_x, n_y)$ is mapped to the pixel positions in the image according to the formula(5)-(6) (Fig 5). This scheme restricts the robot's grasping position within the border of the candidate grasping area. The grasping posture of the drug-taking robot needs to be determined on the basis of the characteristics of different drugs. The transformation of the grasping posture output from the neural network to the drug-taking robot is computed as follows:

$$x = n_x \frac{(u_{max} - u_{min})}{2} \tag{5}$$

$$y = n_y \frac{(v_{max} - v_{min})}{2} \tag{6}$$

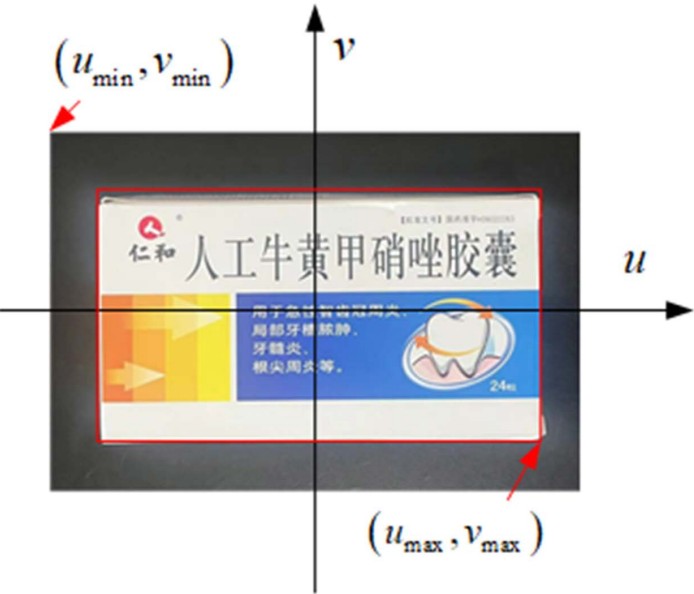

**Fig 5. Schematic of the coordinate system of the robot-grasping action.**

The method mentioned above maps the grasping point position to $(x,y)$. Given that $z$ is calculated using binocular images and that $w$, $h$, and $\theta$ are three parameters obtained on the basis of the characteristics of the medicine box, the final calculated grasping point position is $(x,y,z,w,h,\theta)$. In this research, in the calculation of the grasping pose by using an adsorption-type end effector, the optimal grasping rectangular box is the output in both the length and width directions with respect to the characteristics of the medicine box. The angle between the two directions is 90°. The intersection coverage area of the two directions is calculated to obtain the final adsorption pose. On the basis of the abovementioned mapping action, the position coordinates that the robot's end effector needs to reach can be determined. During the actual grasping process, these position coordinates are sent to the end effector to execute grasping.

## Optimization of the action value function and exploration strategy design

The action value function (Q function) represents the expected reward that a robot can receive when performing a certain action in the current state after reaching the termination state. The Q function is an important evaluation indicator in the Markov decision process and an imperative for judging the quality of a strategy. The strategy gradient method involves the weight coefficient identified before the gradient of the strategy function; the result determines the update direction of the strategy function. The Q function is represented by a neural network called the Q network. According to the basic principles of DRL, the input of the Q network should include the current state and the actions of the robot. The inputs of the Q network for the grasping task are the candidate grasping region image and $(x,y,z,h,\theta)$. The structure of the Q network is shown in Fig 6.

The features extracted by the Q network are the same as those extracted by the policy network. The Q network adopts the same convolutional network as the policy network and shares the parameters of the convolutional network with the policy network during training. The candidate captured region images are not on the same order of magnitude as $(x, y, z, w, h, \theta)$, and their differences are relatively large. In this research, the convolutional layers of the CNN are used to extract image feature information and action values, which are then processed by a fully connected layer with 256

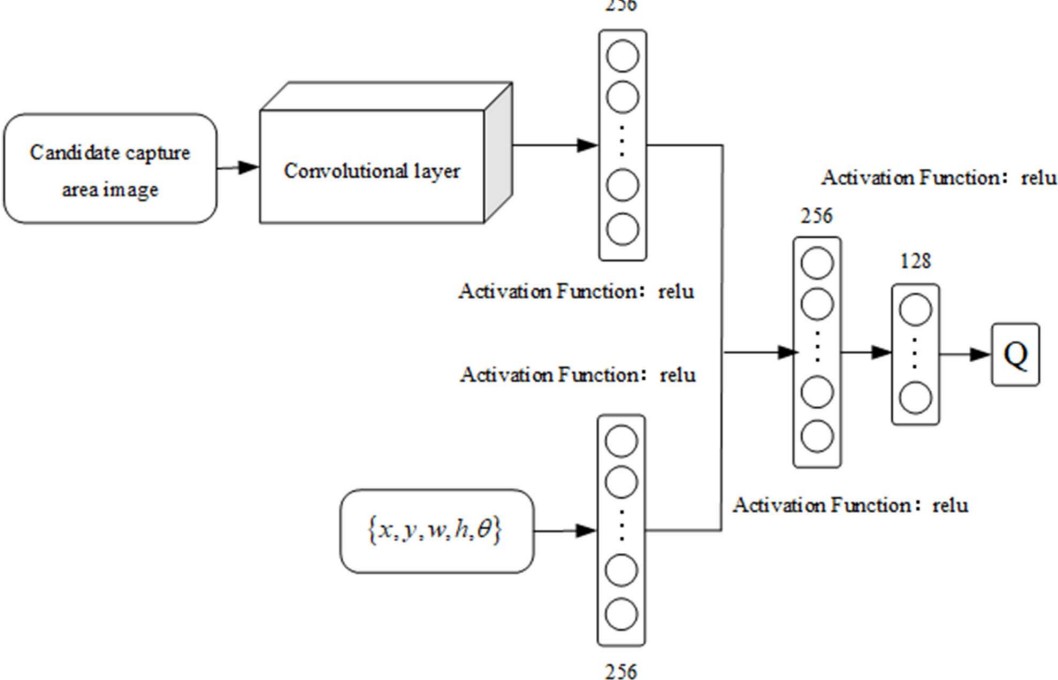

**Fig 6. Schematic of the Q network structure.**

neurons. The data from each part are merged into the same fully connected layer. Here, a fully connected layer with 128 neurons and an output layer with 1 neuron are used.

The DDPG is an offline DRL method that optimizes the Q network. The optimization objective function is given by:

$$\delta_t = r_t + \gamma Q\left(s_{t+1}, \mu\left(s_{t+1}\right)\right) - Q\left(s_t, a_t\right) \tag{7}$$

The Q network serves as both a part of the objective function and an update target; this configuration greatly increases the instability during the model training process. The Q value may also be overestimated, resulting from overfitting and causing the strategy network to learn incorrect information, which affects the final performance of the system. Thus, the Bellman equation of the Q function is used by the T-D algorithm [25] to establish a time difference. This difference is represented by an objective function, which can solve the problem of overestimation in DDPG and limit overfitting of the Q network. In the T-D algorithm, Q-learning [26] is the optimal algorithm used to solve offline learning. When a target network is added, the dual Q-learning algorithm [27] can be used to solve the instability problem in the training process. When the Q network is optimized, the dual Q-learning algorithm can use the target network to calculate the objective function. The network structure is shown in Fig 7.

The task of the Actor is to find A and maximize the output Q:

The target networks Q′ and μ′ for the Q network and policy network, respectively, are set according to Eq. (7). The objective function is calculated as follows:

$$\delta_t = r_t + \gamma Q'\left(s_{t+1}\left|\mu'\left(s_{t+1}\left|\theta_t\right)\right)\right) - Q\left(s_t, a_t\right) \tag{8}$$

The target network of the dual Q-learning algorithm directly copies the various parameters of the original network; however, the original network needs to be updated every step during training. Typically, the target network is updated every

step, with each update directly copying the parameters from the original network. As the target is updated every step, a delay error occurs when a corresponding update is performed in the Q network. The delay error generated in each step of the 0-*N* step increases as the number of steps increases. The target network avoids implementing updates every *N* steps to ensure a uniform delay error; that is, the updates are conducted every step. Compared with the original network, the update of the target network lags behind. The specific update formula is expressed as follows:

$$\theta^{Q'} \leftarrow \tau\theta^{Q} + (1-\tau)\,\theta^{Q'}$$
$$\theta^{\mu'} \leftarrow \tau\theta^{\mu} + (1-\tau)\,\theta^{\mu'}$$

(9)

$\theta^{Q}$, $\theta^{Q'}$, $\theta^{\mu}$, $\theta^{\mu'}$ are the Q network parameter, the Q function target network parameter, the robot strategy network parameter, and the robot strategy target network parameter, respectively.

The learning stage of reinforcement learning corresponds to the autonomous exploration stage of the medication-dispensing robot. This robot explores all workspaces as much as possible and learns the advantages and disadvantages of actions on the basis of the reward function evaluation. As deterministic strategies are selected in this research, the drug retrieval robot repeatedly performs the same drug grasping action in the same state after initializing the strategy parameters and does not have exploratory ability. Random Gaussian white noise $\zeta_t \sim N(0, \sigma^2)$ with a mean of 0 and a variance of $\sigma^2$ is added to the action space for disturbance to enhance the exploration capability of the medication retrieval robot.

$$a_t = \mu\,(t) + N\left(0, \sigma^2\right) \times \theta_t$$

(10)

where the variance $\sigma^2$ controls the exploration space of the medication retrieval robot, $\theta_t = \widetilde{\theta} - \left(\widetilde{\theta} - \theta\right)$ is the strategy network parameter after the addition of noise, $\widetilde{\theta}$ is the parameter after disturbance of the strategy network, and $\theta$ is the parameter before disturbance. The Gaussian noise mean is 0. Thus, the best grasping action and actions far from the best grasping action, which is influenced by variance, have high probabilities of being selected. The strategy after the disturbance is denoted by $\widetilde{\mu}$.

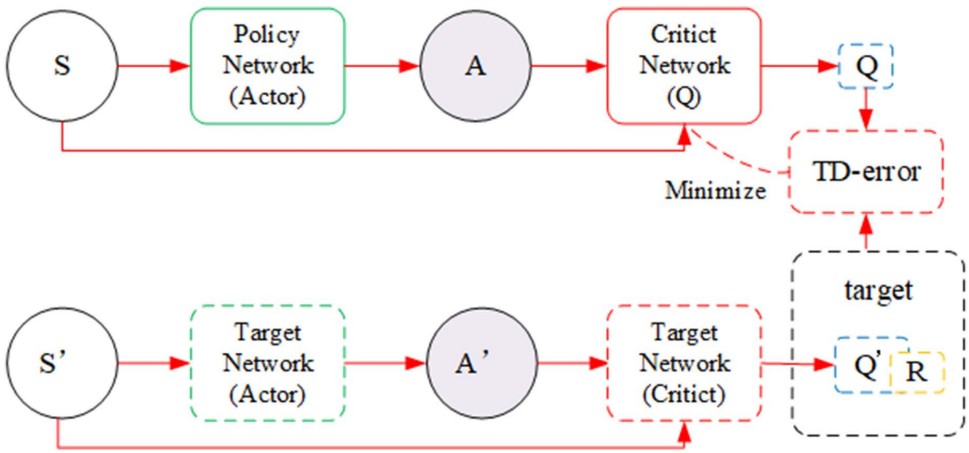

**Fig 7. Double Q-learning network structure diagram.**

The variance, which can be difficult to adjust manually under the influence of noise, must also be considered. Therefore, an adaptive adjustment method is adopted to measure the distance between strategy $\tilde{\mu}$ corresponding to parameter $\theta_i$ after the addition of noise and the original strategy $\mu$ as a measure of the scale of noise exploration control.

$$d\left(\mu, \tilde{\mu}\right) = \sqrt{\frac{1}{N}\sum_{i=1}^{N} E_s\left[\left(\mu(s)_i - \tilde{\mu}(s)_i\right)^2\right]}$$

(11)

In statistics, under the assumption of normal distribution, about 99.7% of the data will fall within the range of $\pm 3\sigma$. The standard deviation reflects the degree of dispersion between the data points and the mean. In the robot grasping space, the direction and range of the spatial set are determined by the standard deviation.The threshold $\delta \epsilon R$ is set as the standard deviation. A distance metric that is less than the threshold indicates insufficient disturbance, which increases the variance. The variance is decreased to obtain the updated formula for variance.

$$\sigma_{k+1} = \begin{cases} \alpha \cdot \sigma_k & d\left(\mu, \tilde{\mu}\right) < \delta \\ \frac{1}{\alpha} \cdot \sigma_k & d\left(\mu, \tilde{\mu}\right) > \delta \end{cases}$$

(12)

where $\alpha = 1.01$ is the influencing factor.

The structural diagram of the autonomous learning DDPG grasping algorithm for the drug-dispensing robot is shown in Fig 8, where $f$ represents the mapping of actions and poses. The candidate grasping areas and policy network outputs are fused to ensure grasping actions by the robot in the candidate grasping areas. The attributes representing the robot's state before executing the action, grasping actions, exploratory learning, and reward values are stored in the experience pool. Once the data in the experience pool reach a certain amount, the Q network can be trained from the experience pool according to uniformly sampled data. After training the Q network, the deterministic gradient method is used to update the robot human control strategy, and this process is repeated until convergence.

The pseudocode for the self-learning DDPG algorithm process is shown in Table 1. *Episode* represents the number of training steps for reinforcement learning, and *T* is the duration sequence of the strategy in each scenario. The duration

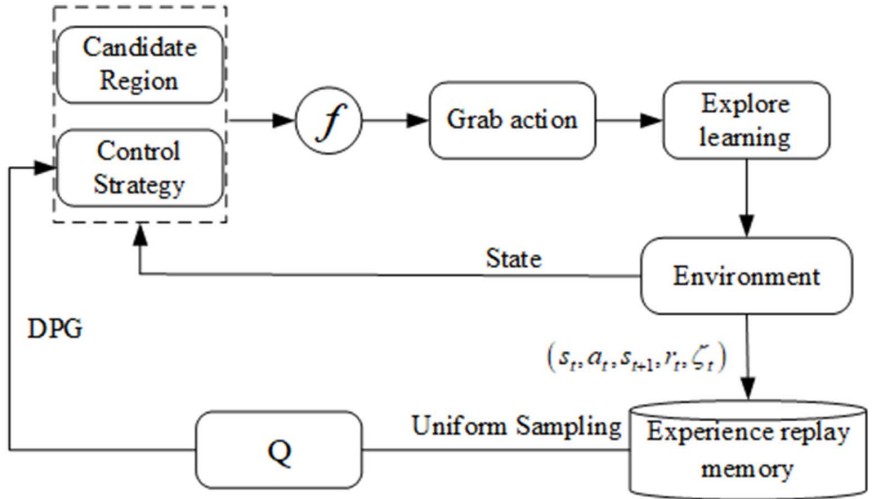

**Fig 8. Block diagram of the self-learning DDPG grabbing algorithm.**

**Table 1. Self-learning DDPG grabbing pseudocode for medicine-taking robots on the basis of detection constraints.**

**Algorithm: Self-learning DDPG pseudocode grasping for medication retrieval robot based on detection constraints**

Initialization: Q network $Q\left(s, a\,\middle|\,\theta^Q\right)$ and grasping strategy network $\mu\left(s, a|\theta^\mu\right)$ parameters $\theta^Q$ and $\theta^\mu$, target networks $Q'$ and $\mu' : \theta^{Q'} \leftarrow \theta^Q, \theta^{\mu'} \leftarrow \theta^\mu$, noise distribution standard deviation $\sigma$, experience pool size $R$

1: *for episode* = 1, *M do*

2: Get current status *s*

3: *for t* = 1, *T do*

4: Obtain actions through the current grasping strategy and exploring noise: $a_t = \mu\left(s_t\,\middle|\,\theta^\mu\right) + \zeta_t$

5: Execute action $a_t$ to earn reward value $r_t$ and obtain the next state $s_{t+1}$

6: Store tuple $(s_t, a_t, s_{t+1}, r_t, \zeta_t)$ in the experience pool

7: Calculate metric distance $d\left(\mu, \tilde{\mu}\right)$, variance $\sigma$, and strategy network parameter $\theta_t$

8: Sample N tuples $(s_t, a_t, s_{t+1}, r_t, \zeta_t)$ from the experience pool through uniform sampling

9: Calculate $y_i = r_i + \gamma Q'\left(s_{i+1}, \mu'\left(s_{i+1}\,\middle|\,\theta^{\mu'}\right)\middle|\theta^{Q'}\right)$

10: Using Adam to optimize Q network parameters and minimize loss values: $L = \frac{1}{N}\sum_i \left(y_i - Q\left(s_i, a_i\,\middle|\,\theta^Q\right)\right)^2$

11: Using deterministic policy gradient method to update policy network parameters:

$\nabla_{\theta^\mu} J \approx \frac{1}{N}\sum_i \nabla_a Q\left(s, a\,\middle|\,\theta^Q\right)\Big|_{s=s_i, a=\mu(s_i)} \nabla_{\theta^\mu}\mu\left(s\,\middle|\,\theta^\mu\right)\Big|_s$

12: Update target network parameters:

$\theta^{Q'} \leftarrow \tau\theta^Q + (1-\tau)\theta^{Q'}$

$\theta^{\mu'} \leftarrow \tau\theta^\mu + (1-\tau)\theta^{\mu'}$

13: *end for*

14: *end for*

sequence captured by the medication retrieval robot is set to 1. During initialization, the Xavier initialization method [28] is selected, and the Adam optimizer is used by the neural network [29]. The Adam algorithm applies default hyperparameters, namely, a learning rate α of 0.001, decay rates of $\beta_1$=0.9 and $\beta_2 = 0.999$, and a setting of $\varepsilon = 10^{-8}$, to maintain numerical stability.

## Experiments and results

The self-learning DDPG grasping algorithm includes two parts. First, the calculated grasping point pose is output via the strategy network, which is known as grasping point detection and serves as the input of the Q network. Second, the robot's autonomous grasping ability is trained via reinforcement learning. The experiments performed in this study were validated from two aspects. First, the accuracy of the strategy network's grasping point detection was verified on a public dataset and applied to actual drug-grasping point detection. Second, the success rate of the robot's autonomous learning and grasping was simulated and calculated for the autonomous learning network.

### Verification and analysis of the grasping point detection results

(1) **Standard dataset capture point detection verification.** The grasping dataset [30] of Cornell University was selected for the experiment, from which the accuracy of the output grasping point pose of the strategy network model was verified. The Cornell dataset includes 1035 images, which cover 280 different everyday objects. Fig 9 shows a section of this dataset. In the captured dataset, the precaptured poses are represented by the captured rectangular boxes labeled in the images. The images are used as network inputs, whereas the labeled captured rectangular boxes are used as outputs. The captured rectangular boxes in this section are output only in a single direction. After training, the proposed strategy network can successfully detect the captured rectangular boxes suitable for capturing the target object from the images.

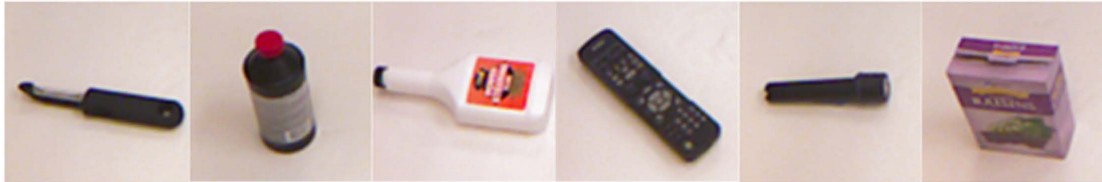

**Fig 9. Cornell crawled dataset.**

Point detection verification is performed on some images in the Cornell dataset via the proposed strategy network. The results of the grasping detection are shown in Fig 10. The first row represents the image input of the model, and the second and third rows represent the outputs of the model. The second row is represented by a schematic of the reference rectangular box captured at each position, with all available positions marked to select the most suitable position for capture. The network can find suitable positions for grasping the target object from the image. For some target objects with obvious handles, the network can output predicted grasping rectangles near the handles. The third row of images displays the highest scoring predicted grasping rectangles, which can sufficiently reflect the pose suitable for grasping the target.

Table 2 lists the detection accuracy results of the mainstream methods and the proposed model on the Cornell dataset. The final performance metrics are calculated as follows, select 50 objects in the dataset, detect each object 10 times, and calculate the average grasping detection accuracy ± standard deviation. The grasping detection accuracy of the proposed method is high, indicating the advantages of self-learning strategy networks. Attribute gains to candidate region extraction (reduced search space) and adaptive DDPG exploration. Although the running speed of the proposed model is slightly lower than that of Redomon, its accuracy is improved by 15%. Compared with other methods, the proposed model performs better in terms of both capture detection accuracy and running speed, the standard deviation is also the smallest.

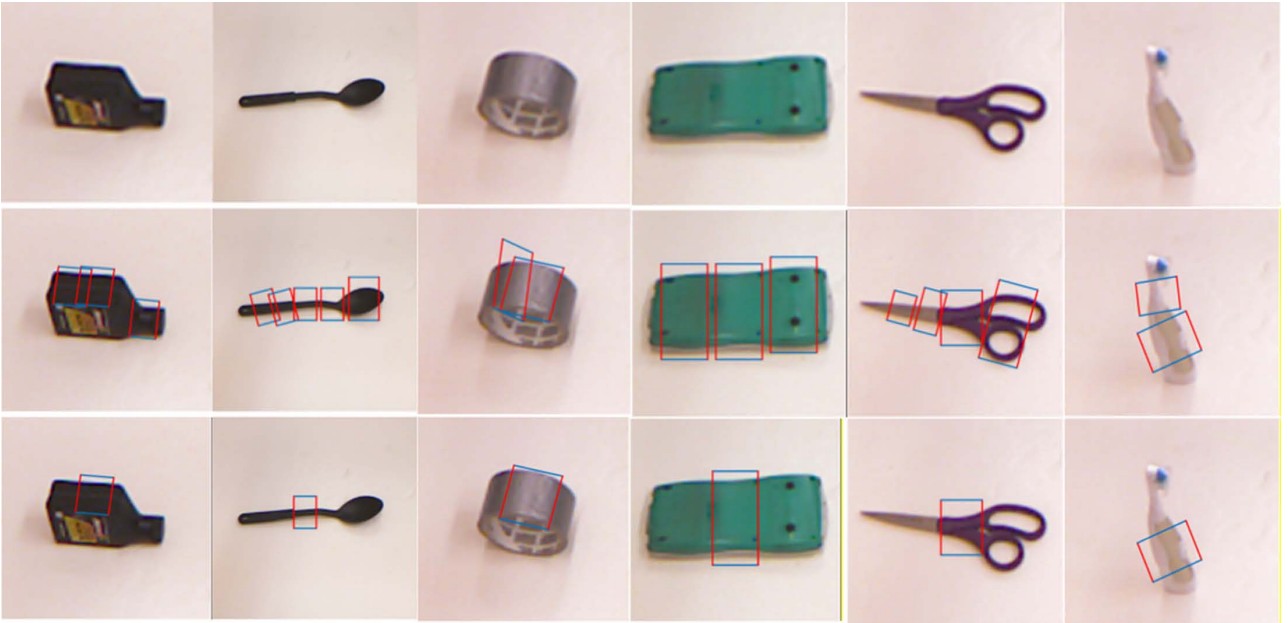

**Fig 10. The results of the grasping detection.**

**Table 2. Grab detection accuracy and running time of different methods.**

| Method | Detection accuracy | Times |
|---|---|---|
| Luca [31] | 72.9%±4.7 | 15.7s |
| Lenz | 73.9%±3.9 | 14.5s |
| Redomon [32] | 86%±3.6 | 0.86s |
| Kim [33] | 76.3%±4.5 | 17.6s |
| Ours | 91.7%±2.3 | 0.9s |

**(2) Actual drug-grasping point detection results.** The accuracy of the strategy network model for drug-grasping point detection calculation was evaluated by subjecting actual drugs to grasping point calculation verification, such as single-drug-grasping point detection and multidrug-grasping point detection. The images of the candidate grasping region are used as the test images for the proposed network model. All possible grasping reference rectangular boxes were considered. The results of grasping point detection are shown in Fig 11. The image input is represented by the first-row

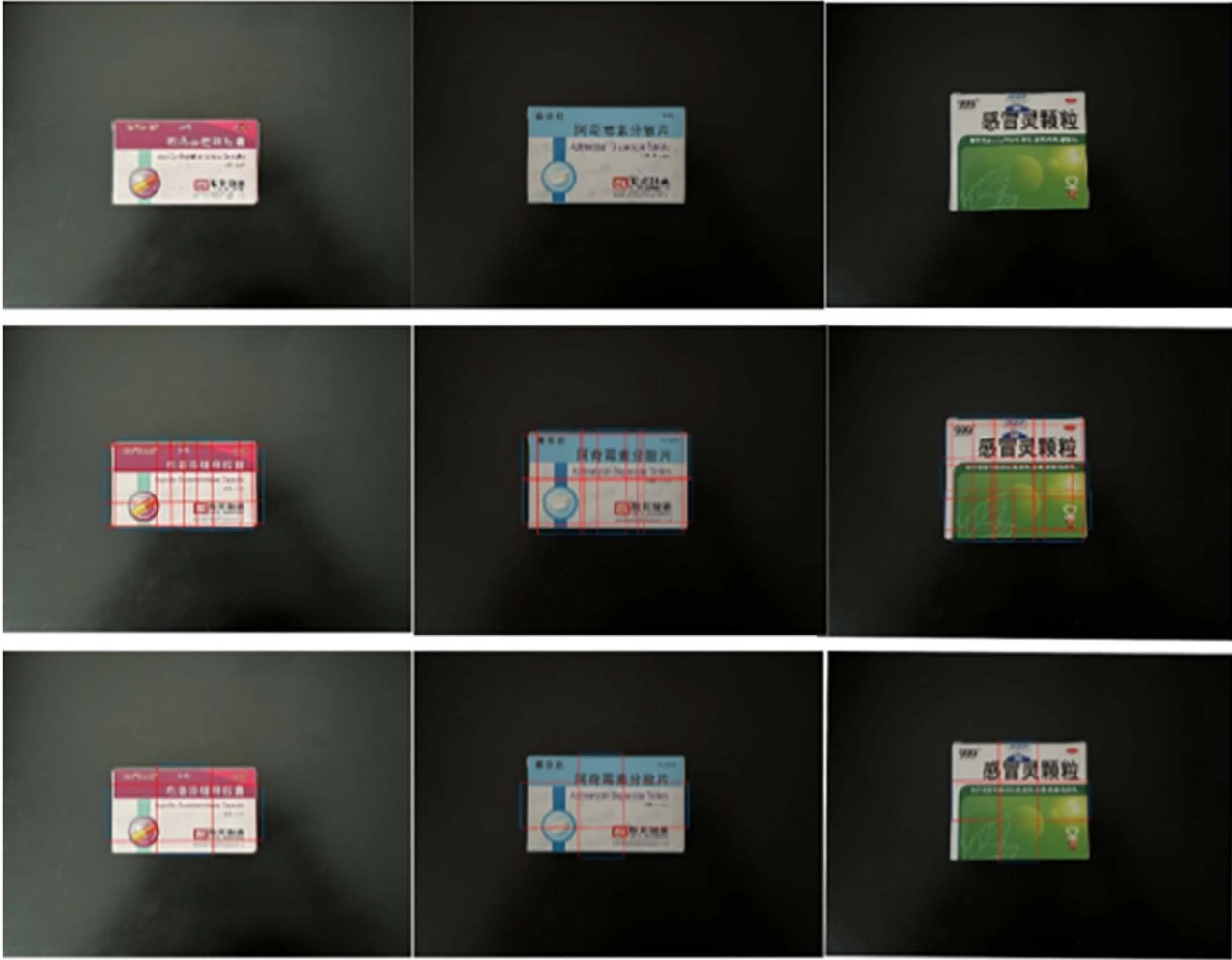

**Fig 11. The results of grasping point detection.**

model, whereas the output is represented by the second and third rows. The second row corresponds to a schematic of all graspable position reference rectangles. The third line of the image shows the final predicted position of the captured rectangular box. The characteristics of the medicine box (i.e., the grasping rectangular box for optimal prediction) suggest the need to determine the optimal positions in both the horizontal and vertical directions. The proposed method can find a suitable position for grasping the medicine from the image, and the predicted grasping rectangular box can sufficiently reflect the posture suitable for grasping the medicine. For the grasping posture of the adsorption end effector, the center coordinates of the overlapping area in both directions correspond to the optimal grasping detection points. During testing, the network utilizes GPU computing, with a detection rate of up to 70 frames per second, which can meet the real-time grasping needs of robots.

Table 3 lists the statistical results of the accuracy of the investigated mainstream methods and the network model strategy proposed in this study for detecting the grasping points of a single drug. The final performance metrics are calculated as follows, select three drugs, twenty calculations were performed for each drug, and determine the average capture detection accuracy ± standard deviation. Compared with the other four methods, the proposed network model has the highest accuracy in detecting drug-grasping points. Although the running speed of the proposed algorithm approximates that of Redomon's method, it is higher than those of the other three algorithms, The standard deviation is also the smallest. The proposed algorithm demonstrates the good performance of the network model.

Grasping point detection calculations were performed in the presence of multiple drugs to further verify the generalizability of the model. The visualization effect is shown in Fig 12. The first image shows the input image, The second image shows all the detectable grasping poses, and The third image shows the optimal grasping pose. The recognition results reveal that the proposed algorithm can identify the position information of each target drug in the image and accurately calculate the grasping pose of each drug. The proposed algorithm has also shown excellent performance in calculating the grasping points of multiple target drugs.

**Table 3. Grab detection accuracy and running time of different methods.**

| Method | Drug 1 | | Drug 2 | | Drug 3 | |
|---|---|---|---|---|---|---|
| | Detection accuracy | Times | Detection accuracy | Times | Detection accuracy | Times |
| Luca | 69.3%±5.2 | 18.1s | 72.6%±5.1 | 15.6s | 74.3%±5.1 | 17.9s |
| Lenz | 72.1%±3.9 | 14.5s | 73.1%±3.5 | 13.5s | 74.7%±3.7 | 15.2s |
| Redomon | 82.9%±2.8 | 1.2s | 83%±2.6 | 1.3s | 83.9%±2.6 | 1.5s |
| Kim | 71.2%±3.7 | 15.7s | 72.6%±3.7 | 16.9s | 71.5%±3.7 | 18.3s |
| Ours | 90.6%±0.0 | 1.1s | 92.8%±2.1 | 1.3s | 92.3%±0.0 | 1.3s |

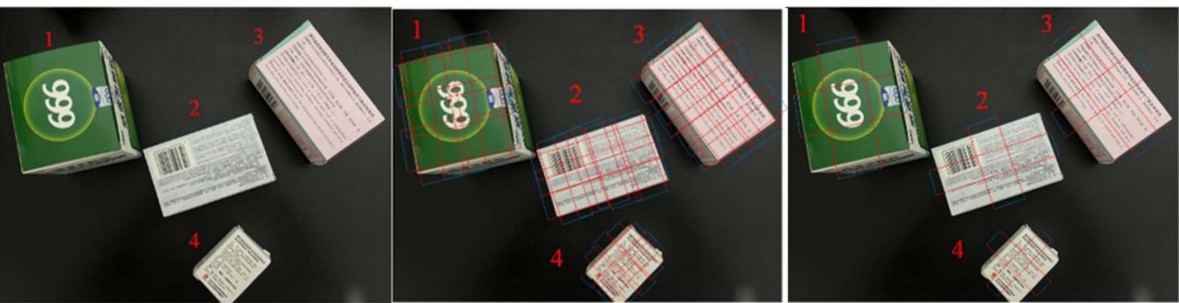

**Fig 12. The visualization effect.**

Table 4 lists several methods and the proposed model and their ability to calculate the accuracy and running time of grasping point detection for multiple target drugs. Twenty grasping detections for each of the four drugs were observed. Calculate the average grasping detection accuracy±standard deviation. The data represent the statistical results of multiple grasping point detection calculations. Compared with the other four methods, the proposed method has the highest accuracy in detecting the grasping points of drugs. For drugs 1 and 2, the detection accuracies exceeded about 80%. Drug 4 has a relatively small medicine box, and the detection accuracy reached 75%, the standard deviation is also the smallest. In terms of running time, the proposed approach has the shortest running time among several methods for testing multiple drugs.

## Self-directed learning to capture simulation results

A simple robot model was established in a simulation environment to verify the superiority of the self-learning DDPG algorithm over the traditional DDPG algorithms in terms of learning efficiency. The abovementioned methods were used to learn how to grasp rectangular prisms. Learning to grasp rectangular prisms is necessary because the drugs captured in this study were designed mainly for boxed drugs, and the medicine box could be analogized to a rectangular prism (Fig 13). During the grasping simulation, the position and posture of the rectangular prism were constantly changing. Thus, the output of the strategy network was considered, and the Q network was used to evaluate the quality of the strategy network model. Force feedback during grasping was used to determine whether an object would be successfully grasped. In this scheme, if the feedback force exceeds the threshold, then the robot has successfully grasped, and a

Table 4. Grab detection accuracy and running time of different methods.

| Method | Drug 1 | | Drug 2 | | Drug 3 | | Drug 4 | |
|---|---|---|---|---|---|---|---|---|
| | Detection accuracy | Times | Detection accuracy | Times | Detection accuracy | Times | Detection accuracy | Times |
| Luca | 67.3%±4.2 | 15.3s | 58.6%±4.4 | 16.3s | 61.7%±4.3 | 15.7s | 53.4%±6.4 | 16s |
| Lenz | 72.4%±3.9 | 16.2s | 62.4%±3.9 | 16.8s | 64.1%±3.8 | 15.9s | 58.9%±6.1 | 16.8s |
| Redomon | 80.7%±3.5 | 1.5s | 73.7%±4.3 | 2.9s | 76.8%±3.5 | 1.7s | 68.2%±5,2 | 1.7s |
| Kim | 72.2%±3.8 | 11.5s | 64.2%±4.3 | 12.3s | 68.6%±3.8 | 11.8s | 60.1%±5.8 | 12.7s |
| Ours | 87.9%±2.1 | 1.3s | 79.8%±2.8 | 2s | 82.2%±2.7 | 1.5s | 76.9%±2.8 | 1.3s |

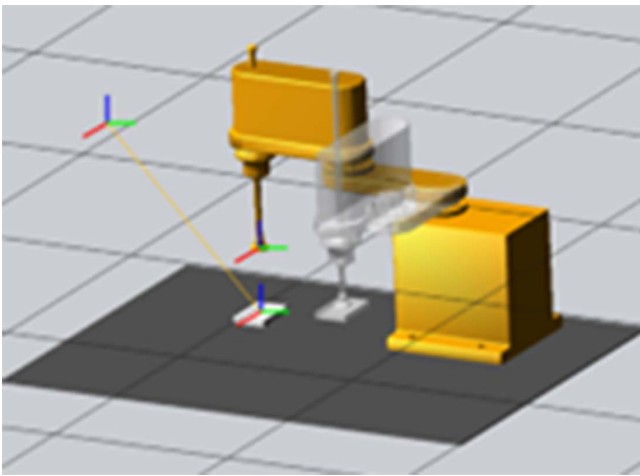

Fig 13. Schematic of the grab simulation.

   

reward value of 1 is given. If the feedback force is less than the threshold, then the robot has failed to grasp, and a penalty value of −1 is given.

During training, the traditional DDPG algorithm、Twin Delay DDPG(TD3)、Soft Actor-Critic(SAC)and the self-learning DDPG algorithm performed similarly, but they differed in terms of the sampling method. The four algorithms shared the same grasping strategy network, action value function, and exploration strategy. They were also trained at 20000 steps each, and their performance metrics were evaluated using the grasping success rate indicator. The final experimental results are shown in Fig 14. Fig 15 shows the results of several algorithms within a 95% confidence interval.

The experimental results show that the model basically converges after 20,000 training steps. Compared with the convergence speeds of the four algorithms, the proposed algorithm outperforms the SAC algorithm, while the DDPG algorithm has the slowest convergence speed. In terms of grasping success rate, the proposed algorithm also achieves a higher success rate than the other three algorithms. This simulation experiment verifies that the self-learning DDPG algorithm has a higher learning speed than the other algorithms do in terms of grasping tasks, thereby achieving the goal of improving the learning speed of the DDPG algorithm via self-learning.

### Actual drug retrieval

To verify the effectiveness of this method in real-world robotic grasping, a set of experimental tasks for a medicine-fetching robot to grasp medicines in real-world scenarios was established. It utilized a vision system in conjunction with the medicine-fetching robot. During the experiment, the drug is placed beneath the visual system, which allows the camera to capture images of the current experimental scene in real-time. The drug detection algorithm efficiently extracts drug

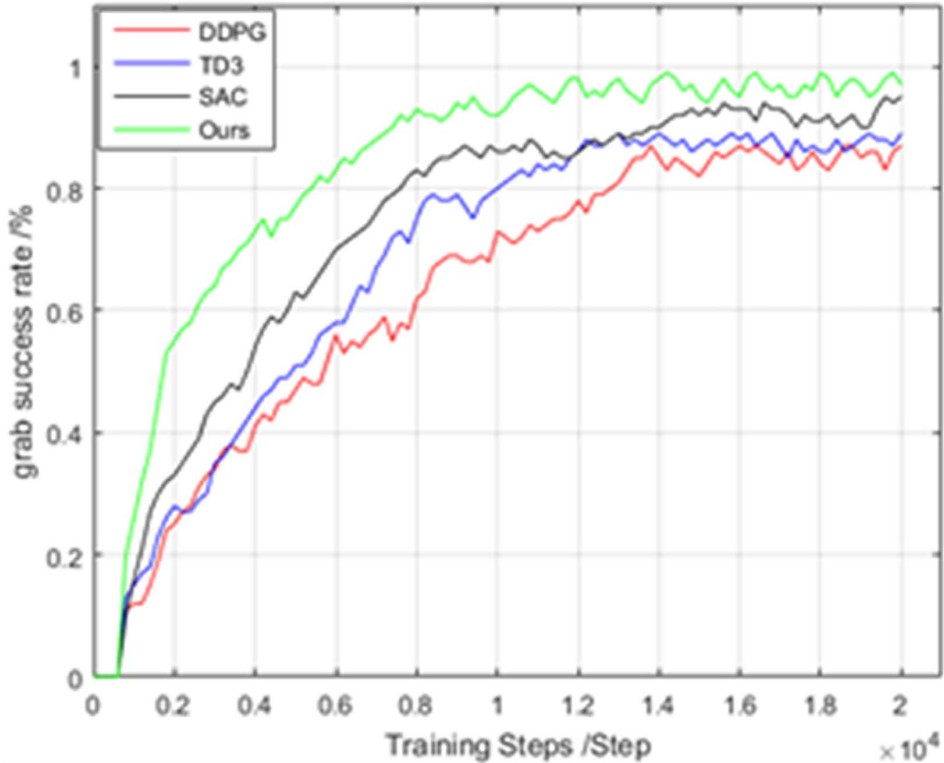

**Fig 14. Grab success rates of the two algorithms under the same number of training steps.**

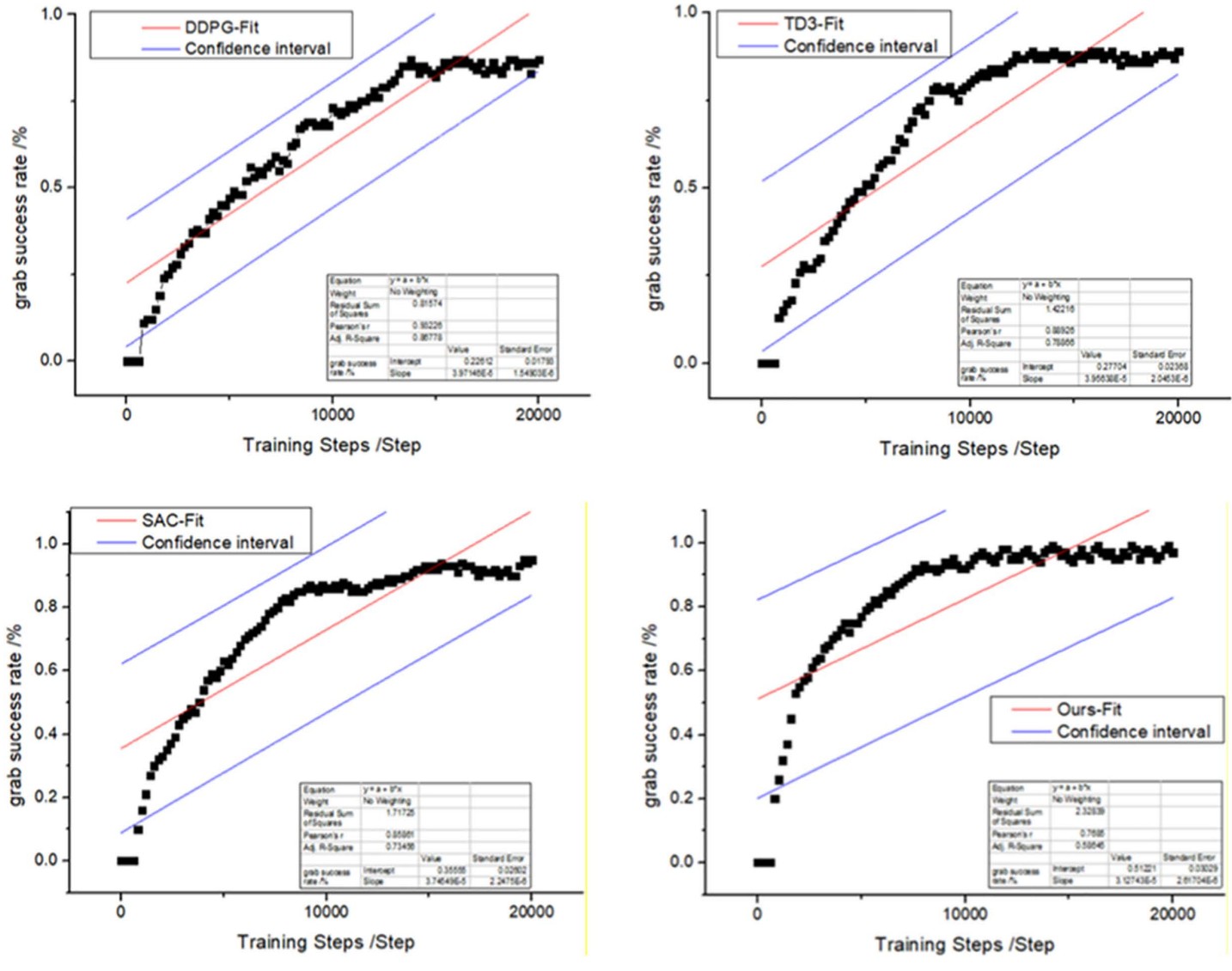

**Fig 15. The performance of four algorithms with 95% confidence intervals.**

information from the image data stream. Then, it locates the captured drug in the current scene. Next, it generates a candidate grasping area where the drug is located. Finally, it utilizes the self-learning DDPG end-to-end neural network to optimize the grasping pose of the drug within the candidate area in real time. Based on the selected end effector, vertical grasping is adopted, where the end effector is placed above the drug with the z-axis direction perpendicular to the operating plane. The grasping position is determined by the overlapping area of the rectangular boxes representing the drug pose in two directions, which controls the robot to reach the pre-grasping pose for grasping. After executing the end stop action, the robot is controlled to move to the designated placement position, complete the drug grasping operation, and wait to return to the initial pose. The motion process of the robot from preparation to grasping and finally placing the medicine in Fig 16.

To compare the applicability of the positioning and grasping algorithm presented in this study for medicine-fetching robots, this chapter contrasts the proposed method with two methods proposed by Guo and Johns [34] for guiding and

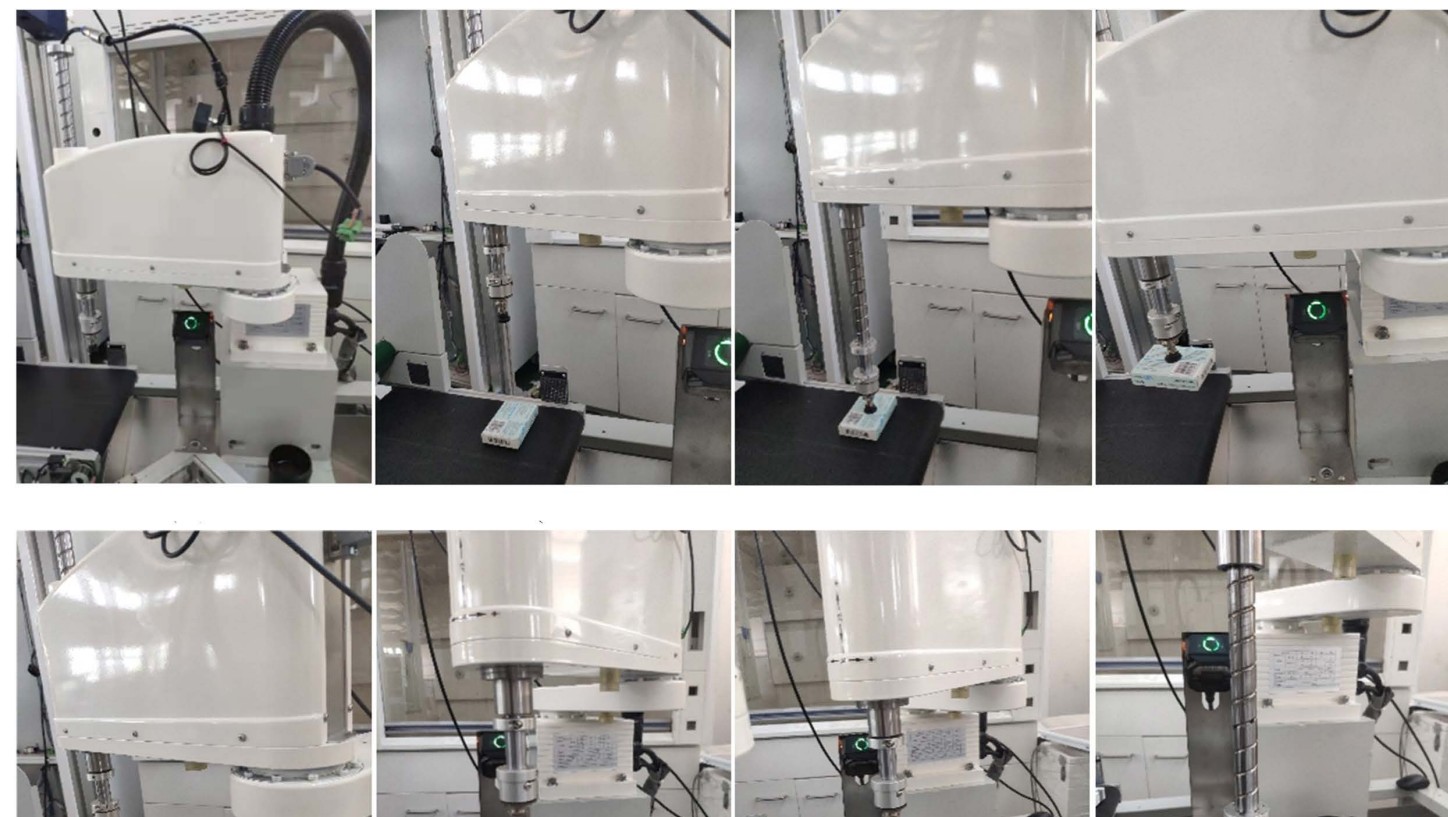

**Fig 16. The process of robot positioning and grabbing medicine.**

grasping medicine-fetching robots. The three methods were tested on single and multiple medicine targets separately. In the single-target medicine grasping experiment, a target medicine was placed on the medicine conveyor belt, and the three methods were used to repeatedly grasp the medicine for 30 times. The experimental statistics for the detection accuracy and grasping success rate± standard deviation. of the medicine are shown in Table 5, where the data are the average values calculated from 30 trials.

The robot was instructed to grasp multiple drugs instead of a single one to verify the applicability of the detection algorithm. The robot was repeatedly used to grasp drugs 30 times, and the statistical results of the detection accuracy and grasping success rate± standard deviation. are presented in Table 6.

Finally, the drugs grasped by the robot were transformed into two identical targets to verify the applicability of the detection algorithm. The drugs were grasped 30 times, and the statistical results of the detection accuracy and grasping success rate± standard deviation. of the drugs are presented in Table 7.

The experimental data presented in Tables 5-3 to 5-5 demonstrate that the monitoring accuracy and grasping success rate of the robot grasping algorithm proposed in this study are relatively high. Therefore, it can handle the actual operation of

**Table 5. Different methods capture statistical results for four single-target drugs.**

|  | Johns | | Guo | | Ours | |
|---|---|---|---|---|---|---|
|  | Detection accuracy | Grasping accuracy | Detection accuracy | Grasping accuracy | Detection accuracy | Grasping accuracy |
| Drug 1 | 86.7%±2.3 | 73.3%±3.4 | 83.3%±3.2 | 80%±3.2 | 96.7%±0.0 | 90%±1.2 |
| Drug 2 | 76.7%±6.4 | 66.7%±5.6 | 76.7%±5.9 | 70%±5.9 | 93.3%±0.0 | 83.3%±1.2 |
| Drug 3 | 80%±2.3 | 73.3%±2.3 | 80%±3.5 | 76.7%±4.5 | 93.3%±0.0 | 86.7%±1.2 |
| Drug 4 | 66.7%±5.6 | 63.3%±5.6 | 73.3%±5.6 | 66.7%±5.9 | 90%±1.6 | 83.3%±4.0 |

**Table 6. Statistical results of multi-target drug detection and capture by different methods.**

|  | Johns | | Guo | | Ours | |
|---|---|---|---|---|---|---|
|  | Detection accuracy | Grasping accuracy | Detection accuracy | Grasping accuracy | Detection accuracy | Grasping accuracy |
| Drug 1 | 80%±4.5 | 70%±5.6 | 83.3%±3.6 | 73.3%±4,7 | 90%±0.0 | 83.3%±4.2 |
| Drug 2 | 66.7%±6.7 | 60%±6.7 | 70%±5.8 | 60%±6.5 | 86.7%±1.2 | 80%±2.8 |
| Drug 3 | 76.7%±7.3 | 66.7%±6.5 | 76.7%±5.8 | 70%±5.9 | 83.3%±2.1 | 80%±2.8 |
| Drug 4 | 66.7%±6.7 | 56.7%±9.8 | 70%±5.8 | 60%±6.5 | 80%±2.1 | 73.3%±4.0 |

**Table 7. Statistical results of two identical drug detection and capture by different methods.**

|  | Johns | | Guo | | Ours | |
|---|---|---|---|---|---|---|
|  | Detection accuracy | Grasping accuracy | Detection accuracy | Grasping accuracy | Detection accuracy | Grasping accuracy |
| Drug 1 | 83.3%±4.5 | 70%±3.7 | 80%±3.9 | 73.3%±6.1 | 96.7%±0.0 | 90%±0.0 |
| Drug 2 | 86.7%±3.8 | 73.3%±3.7 | 83.3%±4.7 | 70%±6.1 | 93.3%±1.8 | 90%±0.0 |

medicine retrieval and dispensing in automated pharmacies. In addition, it exhibits applicability in recognizing and grasping single categories, single drugs, multiple categories, multiple drugs within the same category, and drugs in arbitrary poses.

## Conclusions

A method for point detection via DDPG autonomous learning was proposed on the basis of penalty constraints and DRL. First, drawing on the powerful image feature extraction ability of deep CNNs, a strategy network for the drug retrieval robot to grasp drugs was designed, and the calculated grasping point pose was output via the strategy network. Second, reinforcement learning was combined with autonomous learning, allowing the robot's autonomous grasping ability to be trained. The experiment verified the accuracy of the strategy network's point detection from two aspects: as applied to a public dataset and as applied to actual drug point detection. The accuracy and time of point detection of the proposed method were better than those of the other algorithms. Finally, simulation calculations were conducted on the autonomous learning network to verify the success rate of robot autonomous learning and grasping. Compared with the traditional DDPG algorithm, the algorithm proposed in this study has a higher learning speed and can achieve the goal of improving the learning speed of the DDPG algorithm via autonomous learning.

## Supporting information

**S1 Table. Relevant data underlying the findings described in Fig 14.**
(XLSX)

**S1 Grasp. Relevant codes underlying the findings described in manuscript.**
(TXT)

## Author contributions

**Data curation:** Xiaowen Zhang.

**Formal analysis:** Xiaowen Zhang.

**Funding acquisition:** Xiaowen Zhang.

**Investigation:** Xiaowen Zhang.

**Software:** Tiegang Lv.

**Supervision:** Tiegang Lv.

**Validation:** Tiegang Lv.

**Visualization:** Tiegang Lv.

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
