## [Decision Letter · Decision Letter 0]

25 Jul 2025

Dear Dr. Zhang,

Thank you for submitting your manuscript to PLOS ONE. After careful consideration, we feel that it has merit but does not fully meet PLOS ONE’s publication criteria as it currently stands. Therefore, we invite you to submit a revised version of the manuscript that addresses the points raised during the review process.

**ACADEMIC EDITOR:** The manuscript presents many major concerns as highlighted by both reviewers. However, I see merits on the work. Please, address all the concerns rised by the Experts and provide an updated version of the manuscript.

We look forward to receiving your revised manuscript.

Kind regards,

Andrea Tigrini, Ph.D.

Academic Editor

PLOS ONE

Journal Requirements:

“This work was supported by the Youth Project of Shanxi Province Basic Research Program (Free Exploration Category, 202303021222299).”

Additional Editor Comments (if provided):

The manuscript presents many major concerns as highlighted by both reviewers. However, I see merits on the work. Please, address all the concerns rised by the Experts and provide an updated version of the manuscript.

Reviewers' comments:

Reviewer's Responses to Questions

**Comments to the Author**

1. Is the manuscript technically sound, and do the data support the conclusions?

Reviewer #1: Partly

Reviewer #2: No

2. Has the statistical analysis been performed appropriately and rigorously?

Reviewer #1: No

Reviewer #2: No

3. Have the authors made all data underlying the findings in their manuscript fully available?

Reviewer #1: Yes

Reviewer #2: No

4. Is the manuscript presented in an intelligible fashion and written in standard English?

Reviewer #1: Yes

Reviewer #2: No

Reviewer #1: The manuscript proposes a DDPG-based autonomous learning algorithm for drug retrieval robots, it addresses challenges in grasping accuracy and real-time performance.

While the methodology is interesting, the manuscript requires significant improvements in clarity, technical rigor, and validation to meet publication standards. Below are detailed concern and recommendations.

1. The "detection constraints" (central to the title) are never formally defined. It is not clear how boundaries are analysed (Section 3.1). The constraints must be mathematically defined and how much candidate region extraction boost performance should be evaluated and justified.

2. Equation (8) assumes a pinhole camera model but omits lens distortion correction, critical for real-world accuracy. Please take this into account.

3. The adaptive noise exploration (Eq. 17) lacks justification for threshold δ and decay factor α. Please provide these justifications.

4. "The proposed method has more accurate grasping poses" in the Abstract is not statistically tested. It should be added.

5. Latency (0.9s) is claimed but not benchmarked against robot control cycles. Please compare latency to robot actuator response times (e.g., typical 10–100ms).

6. The autonomous learning component is compared only to vanilla DDPG, not state-of-the-art alternatives (e.g., SAC - Soft Actor-Critic, TD3 - Twin Delayed Deep Deterministic Policy Gradient). Please add alternative comparisons.

7. In the introduction challenging grasping problems such as those showed in the paper “Automating the hand layup process: On the removal of protective films with collaborative robots” https://doi.org/10.1016/j.rcim.2024.102899, should be mentioned.

8. The manuscript reveal a limited real-world testing, experiments use public datasets (Cornell) and simple simulations (rectangular prism). Real robot trials are absent. Test on a physical robot with varied drug geometries (vials, blister packs) is recommended.

9. The DDPG training in Section 4.3.2 uses a simplistic rectangular prism, ignoring real drug variability. Please validate autonomous grasping in a physics simulator with randomized drug poses.

10. Superior accuracy (92.1% vs. 86.4% in Table 2) is presented without technical explanation. Is this due to boundary analysis or DDPG? Please clarify. Please attribute gains to candidate region extraction (reduced search space) and adaptive DDPG exploration.

11. Statistical Reporting in Fig. 12 seems to be inadequate; the success rates (Fig. 12) lack variance measures. Please add confidence intervals to Fig. 12; report mean ± std. dev. in tables.

12. Results are disconnected from real-world pharmacy demands (e.g., FDA standards for medication retrieval accuracy) Please discuss applicability in hospital settings, citing safety margins (e.g., <x% critical=" " drugs=" " error=" " for=" "></x%>

Reviewer #2: The proposed work aimed at providing a method for optimizing autonomous robot grasping for drug retrieval using reinforcement learning, specifically deep deterministic policy gradient (DDPG) algorithm. I have some major concerns that mainly regard the writing and the methodological rigor. My major doubt regards the novelty of the proposed approach: it is not clear if a novel approach was proposed or if an already known approach was applied to the specific problem of drug retrieval. This should have been better clarified throughout the text. Moreover, the description of the performed experiments is lacking, thus hampering a thorough evaluation of the methodological power. Point by point comments are provided in the following.

- Abstract should report some numerical results.

- Aim of the study should be clearly stated and should be more specific about the novelty.

- The Related Work section is not informative and needs to be rewritten. Related Works should report in brief the relevant literature in the field of the deep reinforcement learning strategies and the state of the art and latest relevant methods for the field of drug retrieval with robot grasping. Replicating theoretical formulas already consolidated makes the manuscript very cumbersome.

- In the references section there are no references regarding automatic drug retrieval with robot grasping. This raises concerns about whether the research question is grounded in current developments or unmet needs in the field. Recent work in pharmacy automation and robotic manipulation should be cited to better contextualize the relevance and novelty of the contribution.

- The Methods (section 3), as it is written, do not allow to understand what part of the proposed method is new with respect to the existing literature.

- In section 3.2 it is stated “ nx ny is mapped to the pixel positions in the image according to the formula (Fig. 3).”, but no equation is cited and a figure is mentioned instead. Which “formula” is this sentence referring to?

- In the results section 4.3.1, how was the detection accuracy computed? What does “The images of the candidate grasping region are used as the test images for the proposed network model” mean? How was the dataset of the “(2) Actual drug-grasping point detection results” experiment obtained? Materials used and performed experiments must be carefully explained.

- The basis for the statement “The proposed method can find a suitable position for grasping the medicine from the image, and the predicted grasping rectangular box can sufficiently reflect the posture suitable for grasping the medicine.” is unclear. Is this claim supported by quantitative evaluation? The red lines in Figure 9 are difficult to be interpreted.

- It is not specified why the algorithms used for comparisons where selected: where those algorithms the gold standards? Moreover, are those algorithms/approaches open source? It should be specified if such algorithms where implemented to carry on comparison experiments.

- Formatting of references should be carefully assessed. Reference n 29 is wrongly formatted (i.e., first names are reported instead of last names).

**Do you want your identity to be public for this peer review?** For information about this choice, including consent withdrawal, please see our Privacy Policy

Reviewer #1: No

Reviewer #2: No

---

## [Author Response · Author response to Decision Letter 1]

22 Sep 2025

List of Responses

Dear Editors and Reviewers

On behalf of my co-authors, we thank you very much for giving us an opportunity to revise our manuscript, we appreciate editor and reviewers very much for their positive and constructive comments and suggestions on our manuscript entitled “An algorithm for drug retrieval based on robot-grasping detection constraints and DDPG autonomous learning”. (ID: PONE-D-25-23921).

Those comments are all valuable and very helpful for revising and improving our paper, and they are also having important guiding significance to us researches. We have studied those comments carefully and have made correction which are marked in red in the paper, hoping they will meet the standard. The main corrections and the responds to the reviewer’s comments are as flowing:

Responds to the reviewer’s comments:

Reviewer #1:

1. [Response to comment]:

The "detection constraints" (central to the title) are never formally defined. It is not clear how boundaries are analysed (Section 3.1). The constraints must be mathematically defined and how much candidate region extraction boost performance should be evaluated and justified.

[Responds]:

We sincerely apologize that this part of the content was not detailed enough in the manuscript. In section ‘Candidate capture area generation’, the content of detection constraints and boundary analysis is described in detail, where the location of the drug can be determined through candidate region generation.

2. [Response to comment]:

Equation (8) assumes a pinhole camera model but omits lens distortion correction, critical for real-world accuracy. Please take this into account.

[Responds]:

Considering the reviewer’s suggestion, Equation (8) is Equation (3), we have supplemented the camera calibration process by utilizing the method described in reference [23], which has reduced the error caused by distortion to a certain extent.

3. [Response to comment]:

The adaptive noise exploration (Eq. 17) lacks justification for threshold δ and decay factor α. Please provide these justifications.

[Responds]:

Thank you for reviewer’s suggestion. We apologize for discovering a writing error α=1.01. In statistics, under the assumption of normal distribution, about 99.7% of the data will fall within the range of . The standard deviation reflects the degree of dispersion between the data points and the mean. In the robot grasping space, the direction and range of the spatial set are determined by the standard deviation, so the threshold is chosen as the standard deviation. In terms of implementation, different algorithms have different distance metrics d before and after disturbance. If the distance metric is less than the threshold, it indicates that the disturbance is not sufficient and the search space needs to be increased. On the contrary, if the distance metric is greater than the threshold, it indicates that the disturbance is too large and the search space needs to be reduced.

4. [Response to comment]:

"The proposed method has more accurate grasping poses" in the Abstract is not statistically tested. It should be added.

[Responds]:

According to the reviewer’s suggestion, "The proposed method has more accurate grasping poses" in the Abstract. In the experimental process, the main focus was on statistical description of the capture detection accuracy in different datasets. We have changed it to a description using the capture detection accuracy in the abstract.

5. [Response to comment]:

Latency (0.9s) is claimed but not benchmarked against robot control cycles. Please compare latency to robot actuator response times (e.g., typical 10–100ms).

[Responds]:

In the experimental part, latency (0.9s) refers to the time it takes for the visual system to capture images and output the grasping pose, excluding the average response time of the robot.

6. [Response to comment]:

The autonomous learning component is compared only to vanilla DDPG, not state-of-the-art alternatives (e.g., SAC - Soft Actor-Critic, TD3 - Twin Delayed Deep Deterministic Policy Gradient). Please add alternative comparisons.

[Responds]:

We sincerely apologize that the content didn't describe this part in detail enough in the comparative experiment. In section ‘Self-directed learning to capture simulation results’, we included a comparison with SAC and TD3 algorithms.

7. [Response to comment]:

In the introduction challenging grasping problems such as those showed in the paper “Automating the hand layup process: On the removal of protective films with collaborative robots” https://doi.org/10.1016/j.rcim.2024.102899, should be mentioned.

[Responds]:

Considering the reviewer’s suggestion, we mentioned the challenging grasping problems with this paper in the introduction, and marked in red.

8. [Response to comment]:

The manuscript reveal a limited real-world testing, experiments use public datasets (Cornell) and simple simulations (rectangular prism). Real robot trials are absent. Test on a physical robot with varied drug geometries (vials, blister packs) is recommended.

[Responds]:

In the experimental part, we have added real robot trials, and the target object in the paper is boxed drugs, so the experiment mainly focuses on detecting and grasping boxed drugs. We will consider grasping other varied drug geometries (vials, blister packs) for future work.

9. [Response to comment]:

The DDPG training in Section 4.3.2 uses a simplistic rectangular prism, ignoring real drug variability. Please validate autonomous grasping in a physics simulator with randomized drug poses.

[Responds]:

Due to the fact that our main application in automated pharmacies is boxed drugs, and we also use adsorption end effectors, in order to verify the effectiveness of candidate area boundary analysis and self-learning networks, we only trained rectangular prisms in Section ‘Self-directed learning to capture simulation results’.

10. [Response to comment]:

Superior accuracy (92.1% vs. 86.4% in Table 2) is presented without technical explanation. Is this due to boundary analysis or DDPG? Please clarify. Please attribute gains to candidate region extraction (reduced search space) and adaptive DDPG exploration.

[Responds]:

Considering the reviewer’s suggestion, the detection accuracy in Table 2 is mainly due to the boundary analysis of candidate grasping areas, the input of images into the self-learning detection network, and the final output of the target's grasping detection accuracy. It has been marked in red in the manuscript.

11. [Response to comment]:

Statistical Reporting in Fig. 12 seems to be inadequate; the success rates (Fig. 12) lack variance measures. Please add confidence intervals to Fig. 12; report mean ± std. dev. in tables.

[Responds]:

Thank you for reviewer’s suggestion. We apologize for not describing it enough in the manuscript. In Fig 14, we added confidence intervals and included mean ± std in the table.

12. [Response to comment]:

Results are disconnected from real-world pharmacy demands (e.g., FDA standards for medication retrieval accuracy) Please discuss applicability in hospital settings, citing safety margins (e.g.).

[Responds]:

Thank you for reviewer’s suggestion. Currently, our method is in the testing stage in automated pharmacies and will be further improved based on your valuable suggestion.

Reviewer #2:

1. [Response to comment]:

Abstract should report some numerical results.

[Responds]:

Thank you for reviewer’s suggestion, we did not consider everything comprehensively in the manuscript, and have already included numerical results in the Abstract.

2. [Response to comment]:

Aim of the study should be clearly stated and should be more specific about the novelty.

[Responds]:

Considering the reviewer’s suggestion, we have rewritten the aim of the study, clearly describing the quality of the proposed work, and marked in red in the manuscript.

3. [Response to comment]:

The Related Work section is not informative and needs to be rewritten. Related Works should report in brief the relevant literature in the field of the deep reinforcement learning strategies and the state of the art and latest relevant methods for the field of drug retrieval with robot grasping. Replicating theoretical formulas already consolidated makes the manuscript very cumbersome.

[Responds]:

Considering the reviewer’s suggestion, we have written the Related Work section. Added the relevant paper in the field of the deep reinforcement learning strategies and the state of the art and latest relevant methods for the field of drug retrieval with robot grasping.

4.[ Response to comment]:

In the references section there are no references regarding automatic drug retrieval with robot grasping. This raises concerns about whether the research question is grounded in current developments or unmet needs in the field. Recent work in pharmacy automation and robotic manipulation should be cited to better contextualize the relevance and novelty of the contribution.

[Responds]:

Thank you for reviewer’s suggestion. We have added new reference on the direction of robot grasping for automatic medication retrieval.

5. [Response to comment]:

The Methods (section 3), as it is written, do not allow to understand what part of the proposed method is new with respect to the existing literature.

[Responds]:

Considering the reviewer’s suggestion, we have reiterated the novelty of the proposed method in section ‘Proposed method’.

6. [Response to comment]:

In section 3.2 it is stated “nx ny is mapped to the pixel positions in the image according to the formula (Fig. 3).”, but no equation is cited and a figure is mentioned instead. Which “formula” is this sentence referring to?

[Responds]:

We are very sorry that we did not express this part in detail in the manuscript. Formulas refer to formulas 5 and 6.

7. [Response to comment]:

In the results section 4.3.1, how was the detection accuracy computed? What does “The images of the candidate grasping region are used as the test images for the proposed network model” mean? How was the dataset of the “(2) Actual drug-grasping point detection results” experiment obtained? Materials used and performed experiments must be carefully explained.

[Responds]:

In the manuscript, we have detailed the deep topology, how the parameters are tuned and selected. They are not dependent on images.

(1) In the results section ‘Verification and analysis of the grasping point detection results’, The detection accuracy is calculated using the following formula:

(2) We sincerely apologize for the insufficient description. ‘The images of the candidate grasping region are used as the test images for the proposed network model’ refers to the result image of candidate grasping regions output in section ‘Candidate capture area generation’, which serves as input to the policy network, enabling targeted output of grasping poses.

(3) The actual drug capture dataset is a self-built dataset, which contains 50 real target drugs, with 200 training images established for each drug.

8. [Response to comment]:

The basis for the statement “The proposed method can find a suitable position for grasping the medicine from the image, and the predicted grasping rectangular box can sufficiently reflect the posture suitable for grasping the medicine.” is unclear. Is this claim supported by quantitative evaluation? The red lines in Figure 9 are difficult to be interpreted.

[Responds]:

In the experimental part, using both publicly available and self-built datasets for point detection and validation, the output rectangular box represents the appropriate location for grabbing drugs. The red line in Figure 9 (b) is a schematic diagram of all reference rectangular boxes that can be grabbed. Figure 9 (c) shows the final predicted grasping rectangle box, which is grasped at the center position using an adsorption type end effector.

9. [Response to comment]:

It is not specified why the algorithms used for comparisons where selected: where those algorithms the gold standards? Moreover, are those algorithms/approaches open source? It should be specified if such algorithms where implemented to carry on comparison experiments.

[Responds]:

In the experimental part, the comparison algorithms are not gold standards, but they all have their advantages. [9] is the first application of deep learning networks for grasping point detection. [31] highlights the real-time performance of point detection based on deep learning for grasping. [32] mainly highlights the real-time performance of using neural networks for grasping point detection. [33] mainly highlights the real-time performance of grasping point detection. These methods are all open-source.

10. [Response to comment]:

Formatting of references should be carefully assessed. Reference n 29 is wrongly formatted (i.e., first names are reported instead of last names).

[Responds]:

Thank you for reviewer’s suggestion. We have reconstructed, checked, and revised the reference format.

Thank you for your constructive comments.

We tried our best to improve the manuscript and made some changes in the manuscript, already marked in red. These changes will not influence the content and framework of the paper. We need to show our heartfelt appreciation for Editors/Reviewers’ efforts, and hope that the correction will meet with approval.

Once again, thank you very much for your comments and suggestions.

Yours sincerely,

Zhang XiaoWen

E-mail: 503212590@qq.com

---

## [Decision Letter · Decision Letter 1]

15 Oct 2025

An algorithm for drug retrieval based on robot-grasping detection constraints and DDPG autonomous learning

PONE-D-25-23921R1

Dear Dr. Zhang,

We’re pleased to inform you that your manuscript has been judged scientifically suitable for publication and will be formally accepted for publication once it meets all outstanding technical requirements.

Kind regards,

Andrea Tigrini, Ph.D.

Academic Editor

PLOS ONE

Additional Editor Comments (optional):

Authros addressed all the concerns and the manuscript can be accepted for publication.

Reviewers' comments:

Reviewer's Responses to Questions

**Comments to the Author**

Reviewer #3: All comments have been addressed

Reviewer #4: All comments have been addressed

2. Is the manuscript technically sound, and do the data support the conclusions?

Reviewer #3: Yes

Reviewer #4: Yes

3. Has the statistical analysis been performed appropriately and rigorously?

Reviewer #3: N/A

Reviewer #4: No

4. Have the authors made all data underlying the findings in their manuscript fully available?

Reviewer #3: Yes

Reviewer #4: Yes

5. Is the manuscript presented in an intelligible fashion and written in standard English?

Reviewer #3: Yes

Reviewer #4: Yes

Reviewer #3: Thank you for your detailed and thoughtful responses to reviewer comments. I appreciate the effort you have made to address the concerns raised. You have effectively incorporated the major suggestions.

Overall, I am satisfied with the current version of your manuscript. The revisions have strengthened the work, and in my view, it is now well-prepared for publication.

Wishing you success with your ongoing and future research endeavors.

Best regards

Reviewer #4: Dear Authors,

I have found that you've addressed the feedback thoroughly and explicitly including latency clarification, comparitive baselines for algorithms, real world experiments and clarifications of performance gain. The manuscript is technically sound and clearly structured. The problem statement is thoroughly addressed with the proposed methodology. A statistical analysis of the results provided is acceptable, though it could benefit from a deeper quantitative analysis. Overall, I find the manuscript meeting the necessary standards for PLOS One.

**Do you want your identity to be public for this peer review?** For information about this choice, including consent withdrawal, please see our Privacy Policy

Reviewer #3: No

Reviewer #4: **Yes: ** Akshay Aggarwal

---

## [Editor Report · Acceptance letter]

PONE-D-25-23921R1

PLOS ONE

Dear Dr. Zhang,

I'm pleased to inform you that your manuscript has been deemed suitable for publication in PLOS ONE. Congratulations! Your manuscript is now being handed over to our production team.

Kind regards,

on behalf of

Dr. Andrea Tigrini

Academic Editor

PLOS ONE